# Optimizing Stabilization in Singularly Perturbed Problems with SUPG Scheme

## Abstract

This paper introduces ConvStabNet, a convolutional neural network that predicts optimal stabilization parameters for the Streamline Upwind Petrov Galerkin method (SUPG) stabilization scheme. To enhance the accuracy of SUPG in solving partial differential equations (PDE) with interior and boundary layers, ConvStabNet incorporates a loss function that combines a strong residual component and a cross-wind derivative term. ConvStabNet utilizes a shared parameter scheme, enabling the network to learn the correlations between cell properties and their respective stabilization parameters while effectively managing the parameter space. Comparative evaluations against state-of-the-art neural network solvers based on variational formulations demonstrate the superior performance of ConvStabNet. The results affirm ConvStabNet as a promising approach for accurately predicting stabilization parameters in SUPG, thereby establishing it as an improvement over neural network-based SUPG solvers.

## 1 Introduction

The scalar advection-diffusion equation describes various scalar quantities, including species concentration, temperature, and electron continuity in semiconductors. In such equations, the advection term typically dominates over diffusion, leading to what is known as Singularly Perturbed Partial Differential Equations (SPPDEs) Tobiska & Verfurth (1996). Solving these equations using finite element or finite volume methods poses challenges due to the emergence of boundary and interior layers, often resulting in spurious oscillations in the numerical solution. Researchers have proposed stabilization techniques to overcome this issue and eliminate these undesirable oscillations Yadav & Ganesan (2019; 2021). However, the effectiveness of these stabilization techniques relies on a user-defined parameter called the stabilization parameter ($\tau_K$). In practice, determining an optimal value for $\tau_K$ is difficult, as closed-form formulas are unavailable for most cases. Among the various stabilization techniques proposed in the literature, one widely adopted approach is the Streamline Upwind Petrov-Galerkin (SUPG) technique, discussed by Brooks and Hughes Brooks & Hughes (1982); Hughes et al. (1989). The SUPG technique aims to stabilize the weak form of the given PDE by introducing additional diffusion in the upwind direction. The amount of stabilization can be adjusted by controlling the user-defined stabilization parameter ($\tau_K$). Choosing an appropriate value for $\tau_K$ is crucial, as a high value can excessively smooth out oscillations, while a small value may not effectively eliminate the oscillations. To enhance the accuracy of the SUPG technique for SPPDEs, we propose utilizing a convolutional neural network (CNN) to predict the value of $\tau_K$. This prediction is achieved by minimizing an error function, as proposed in John et al. (2011). By incorporating this approach, we can improve the accuracy of the SUPG technique when applied to SPPDEs. Before delving into the details of the proposed technique, we will review existing deep-learning solvers for PDEs and explore how these solvers can be adapted for SPPDEs.

In recent years, significant research has been on utilizing deep learning for solving partial differential equations (PDEs). Deep learning can be applied in two ways: as a direct PDE solver or as an additional tool alongside traditional PDE-solving techniques such as the finite element method (FEM), finite difference method (FDM), and finite volume method (FVM). When deep learning is used as a PDE solver, the problem is formulated as an optimization problem since labeled data for supervised training is generally unavailable.

One commonly employed approach for solving PDEs using neural networks is the Physics-Informed Neural Network (PINN). PINN was introduced in Raissi et al. (2019). Unlike traditional neural network-based supervised learning, where accurate reference data is available, PINN adopts a data-driven strategy that incorporates the underlying physical laws of the problem to formulate the loss function for training the neural network. PINN provides an approximation of the numerical solution by minimizing the residual of the equation while satisfying the imposed boundary conditions.

To illustrate the concept, let's consider a PDE defined on a bounded domain $\Omega \in \mathbb{R}^d$ for $d = 1, 2$, with the following boundary conditions:

$$
\begin{aligned}
\mathcal{D}(u(x)) &= 0, & x \in \Omega, \\
\mathcal{B}(u(x)) &= f, & x \in \partial\Omega.
\end{aligned}
\tag{1}
$$

In Equation equation 1, $u$ represents the unknown solution, $\partial\Omega$ denotes the domain boundary, $\mathcal{D}$ represents a linear or nonlinear differential operator, and $\mathcal{B}$ denotes the boundary condition of the given PDE (e.g., Dirichlet, Neumann, or Robin boundary condition). In PINN, an approximate solution $\hat{u}$ to the equation is estimated using a feed-forward neural network $M$, which can be represented as follows:

$$
\hat{u} = M(x; \theta).
\tag{2}
$$

Here, $x$ represents the input coordinates, and $\theta$ denotes the parameters of the neural network. The goal is to find the optimal values of $\theta$ that minimize the loss function. The loss function in PINN consists of two components: the mean squared error ($MSE_u$) between the approximate solution $\hat{u}$ and the true solution $u$ at interior points, and the mean squared error ($MSE_f$) between the residual term $f$ and zero at boundary points. The overall loss function can be written as follows:

$$
\begin{aligned}
Loss(\theta) &= MSE_u + MSE_f, \\
MSE_u &= \frac{1}{N_u} \sum_{i=1}^{N_u} |\hat{u}^i - u(x_u^i)|^2, \\
MSE_f &= \frac{1}{N_f} \sum_{i=1}^{N_f} |f(x_f^i)|^2.
\end{aligned}
\tag{3}
$$

In Equation equation 3, $x_u^i$, $x_f^i$ are the spatially collocated points in $\Omega$ and $\partial\Omega$ respectively, $N_f$, $N_u$ is the number of boundary and interior points, respectively. Lossequation 3 is minimized to obtain an optimal $\theta$. While PINN was one of the pioneering neural network architectures introduced for PDE solving, its accuracy had certain limitations. As a result, significant progress has been achieved since the development of PINN. One notable advancement in this field is the introduction of Variational Neural Networks (VarNet) for PDE solutions, as presented in Khodayi-mehr & Zavlanos (2020). In the subsequent section, we will delve into the details of VarNet and its application in solving PDEs.

## 1.1 Variational Neural Networks

Variational Neural Networks (VarNet) for the Solution of Partial Differential Equations were introduced in Khodayi-mehr & Zavlanos (2020). VarNet is a PDE solver based on neural networks that employ a unique loss function relying on the PDE's variational (integral) form, in contrast to the differential form used by Physics-Informed Neural Networks (PINNs). The novel loss function of VarNet effectively approximates the solution by incorporating lower-order derivatives. However, both PINNs and VarNet have limitations when applied to Singularly Perturbed Partial Differential Equations (SPPDEs). We propose incorporating a classical stabilization technique into the solvers' loss function to overcome these limitations and improve the accuracy of neural network-based PDE solvers for SPPDEs.

## 1.2 AI-augmented stabilized FEM (AIStab-FEM)

AI-augmented stabilized FEM (AIStab-FEM) is a technique proposed by Sangeeta et al. in their paper Yadav & Ganesan (2022) to predict a global stabilization parameter $\tau_K$ for 2D Singularly Perturbed Partial Differential Equations (SPPDEs) using neural networks. This approach utilizes the strong form of the residual and incorporates the cross-wind derivative term into the loss function. However, there are some limitations associated with this technique.

One limitation of AIStab-FEM is that it can only predict a global $\tau$ value, which is then localized by dividing it with the norm of gradients of the standard solution. This approach does not provide a truly local solution, so it does not work effectively for SPPDEs with spatially varying equation parameters. Additionally, the normalization step assumes that the stabilization parameter has a solution gradient in the denominator. This assumption is not always ideal and can lead to artificially high values of $\tau_K$ in regions with low gradients. To address these limitations, our proposed method aims to predict the stabilization parameter locally for each cell using convolutional neural network (CNN). By doing so, we can overcome the limitations of AIStab-FEM and provide a more accurate and localized solution for SPPDEs. **Contributions**
The contributions of this research work are outlined below:

- A convolution neural network is introduced for predicting cell-wise stabilization parameters for Streamline Upwind Petrov-Galerkin (SUPG) in two-dimensional Singularly Perturbed Partial Differential Equations (SPPDEs). The CNN architecture is designed to effectively capture spatial dependencies and provide accurate predictions of the stabilization parameters.

- An error functional, based on the cross-wind derivative term, is utilized as the loss function for the proposed CNN, referred to as *ConvStabNet*. The error functional, proposed in John et al. (2011), enables the CNN to learn the optimal stabilization parameters by minimizing the discrepancy between predicted and true values.

- The proposed technique is compared with existing approaches to highlight its effectiveness. The following contemporary ideas are considered for comparison:

  – AIStab-FEM: Ai-Augmented Stabilized Finite Element Method Yadav & Ganesan (2022).
  – VarNet: Variational Neural Networks Kharazmi et al. (2019).
  – SUPG Stabilized Finite Element Method N. & R. (1982).

The structure of the paper is outlined as follows: In Section 2, the necessary mathematical preliminaries are presented to facilitate understanding of the proposed method. The details of the proposed method, including the network architecture and the error functional as the loss function, are elaborated in Section 3. Subsequently, a discussion on two prominent neural network-based PDE solvers is presented. Finally, the paper concludes in Section 5.

## 2 MATHEMATICAL PRELIMINARIES

SPPDEs are a class of differential equations with a small diffusion parameter ($\epsilon > 0$) multiplied with the second order differential term. A small value of $\epsilon$ often induces spurious oscillations in the standard Galerkin solution. For a given bounded domain $\Omega \subset \mathbb{R}^d$, where $d \in \mathbb{N}$, an SPPDE is given as follows:

$$
\begin{aligned}
-\epsilon \Delta u + \mathbf{b} \cdot \nabla u &= f(x), &&\text{in } \Omega \subset \mathbb{R}^d, \\
u &= g, &&\text{on } \partial\Omega,
\end{aligned}
\tag{4}
$$

where $\epsilon > 0$ is the diffusion coefficient, also called a perturbation parameter, $\mathbf{b} = (b_1, b_2)^T$ is the convective velocity, $f \in L^2(\Omega)$ is the external source term, $u$ is the unknown scalar term, g is the known boundary value. For smooth $\mathbf{b}$ and $f(x)$, equation equation 4 has a unique solution. We will consider the convection-dominated problems, that is, $\epsilon \ll |\mathbf{b}|$.

### 2.1 WEAK FORMULATION

We utilize the FEM to solve the equation equation 4. The initial step in FEM involves deriving the weak form of the given equation. To derive the weak form equation of equation 4, we multiply it by a function $v \in V := H_0^1(\Omega)$, integrate it over $\Omega$, and then apply integration by parts. The objective is to find a function $u \in H^1(\Omega)$ that satisfies the following condition for all $v \in V$:

$$
a(u, v) = (f, v).
\tag{5}
$$

Here, the bilinear form $a(\cdot, \cdot) : H^1(\Omega) \times H_0^1(\Omega) \to \mathbb{R}$ and the linear form $f(v) : H_0^1(\Omega) \to \mathbb{R}$ are defined as:

$$a(u, v) = \int_\Omega \epsilon \nabla u \cdot \nabla v \, dx + \int_\Omega \mathbf{b} \cdot \nabla u \, v \, dx,$$

$$f(v) = \int_\Omega f \, v \, dx.$$

Let $\Omega_h$ be an admissible decomposition of $\Omega$ and let $K$ represent a single cell in $\Omega_h$. Let $H^1(\Omega_h) \subset H^1(\Omega)$ and $V_h \subset H_0^1(\Omega)$ be finite-dimensional spaces comprising piece-wise continuous polynomials. The discrete form of the equation reads:

Find $u_h \in H^1(\Omega_h)$ such that for all $v_h \in V_h$ we have

$$a_h(u_h, v_h) = (f, v_h), \tag{6}$$

where

$$a_h(u_h, v_h) := \epsilon(\nabla u_h, \nabla v_h) + (\mathbf{b} \cdot \nabla u_h, v_h).$$

## 2.2 Stabilized weak formulation using SUPG

In the Streamline Upwind Petrov-Galerkin (SUPG) method, a residual term is incorporated into the equation's weak form in the streamline's direction. We define $R(u)$ as the residual of Equation equation 4, given by:

$$R(u) = -\epsilon \Delta u + b \cdot \nabla u - f \tag{7}$$

The term $R(u)$ is added to the discretized weak formulation given in equation equation 6. Now, the modified discrete weak form reads:

Find $u_h \in V_h$ such that:

$$a^{SUPG}(u_h, v_h) = (f, v_h), \tag{8}$$

where

$$a^{SUPG}(u_h, v_h) = \epsilon(\nabla u_h, \nabla v_h) + (\mathbf{b} \cdot \nabla u_h, v_h)$$
$$+ \sum_{K \in \Omega_h} \tau_K(-\epsilon \Delta u_h + \mathbf{b} \cdot \nabla u_h - f_h, \mathbf{b} \cdot \nabla v_h)_K.$$

In this work, we focus on the non-negative stabilization parameter $\tau_K$, which plays a crucial role in determining the quality of the approximated solution. The value of $\tau_K$ is of utmost importance as it can significantly impact the accuracy and behavior of the solution. A large value can result in unexpected smearing, while a small value may fail to remove spurious oscillations.

By leveraging the power of deep learning, we aim to develop a model that can effectively estimate $\tau_K$ based on the given input data. This approach allows us to automate the selection process and improve the overall performance and stability of the solution.

## 2.3 Standard stabilization parameter

In the literature, several expressions exist for the stabilization parameter ($\tau$); the most commonly used relation in one-dimensional problems is provided below:

$$\tau_{std}|_K = \frac{h_K}{2|\mathbf{b}|} \left( \coth(Pe_K) - \frac{1}{Pe_K} \right) \tag{9}$$

Here, $Pe_K = |\mathbf{b}|h/2\epsilon$ is the local Peclet number and $h_K$ is the diameter of the cell $K$. However, the conventional expression for $\tau_{std}$ is insufficient for solving all Singularly Perturbed Partial Differential Equations (SPPDEs), and it lacks extendability to high-dimensional problems.

We propose a novel technique based on convolutional neural networks (CNNs) to address this limitation to predict a more generalizable $\tau_K$ value. This approach aims to achieve a lower numerical error in the solution compared to the standard $\tau_K$ approach.

By leveraging the capabilities of CNNs, we aim to capture intricate patterns and dependencies in the data, enabling us to develop a predictive model that can effectively estimate optimal $\tau$ values. This approach is expected to yield improved accuracy and reliability in solving SPPDEs, particularly in scenarios where the conventional $\tau_{std}$ fails to provide satisfactory results.

## 2.4 ERROR METRICS

We use the following metrics to calculate numerical errors in the solution obtained with the $\tau_K$ predicted from *ConvStabNet*. We use them for comparison against the standard $\tau$(equation equation 9), VarNet, and AIStab-FEM as explained in section 1.

$$L^2\text{-error: } \|e_h\|_0 = \|u_h - u_{exact}\|_{L^2(\Omega)} = \left( \int_\Omega (u_h - u_{exact})^2 dx \right)^{\frac{1}{2}}$$

$$\text{Relative } l^2\text{-error:} \|e_h\|_{0,\ell} = \sum_{i=1}^N \frac{\|u_h(x_i) - u_{exact}(x_i)\|_{0,\ell}}{\|u_{exact}\|_{0,\ell}}, \qquad x_i \in \Omega_h$$

$$H^1\text{-seminorm error: } |e_h|_1 = \|\nabla u_h - \nabla u_{exact})\|_{L^2(\Omega)} = \left( \int_\Omega (\nabla u_h - \nabla u_{exact})^2 dx \right)^{\frac{1}{2}}$$

$$L^\infty\text{-error: } \|e\|_{L^\infty(\Omega)} = \text{ess sup}\{|u_h - u_{exact}| : x \in \Omega\}.$$

Here, $u_{exact}$ is the exact solution.

## 3 PROPOSED METHOD: CONVSTABNET

We introduce ConvStabNet, a convolutional neural network (CNN) designed for predicting the value of $\tau_K$. The algorithmic explanation of ConvStabNet can be found in Algorithm 1, and a schematic representation is depicted in Figure 1. The error indicators inspire the loss function employed by the CNN proposed in John et al. (2011), explicitly utilizing the SUPG stabilized weak form (equation equation 8) of the equation.

$$\tau_K(\theta) = \text{ConvStabNet}_{\theta_t}(\epsilon^K, b_1^K, b_2^K, h^K, \|\nabla u_h^K\|_{0,K})$$

$$Loss(u_h(\theta)) = \sum_{K \in \Omega} (\| -\epsilon \Delta u_h + \mathbf{b} \cdot \nabla u_h - f\|_{0,K}^2 + \|q(|\mathbf{b}^\perp \cdot \nabla u_h|)\|_{0,1,K})$$

$$\text{where, } q(s) = \begin{cases} \sqrt{s} & s > 1 \\ 2.5s^2 - 1.5s^3 & \text{otherwise} \end{cases} \tag{10}$$

$$\text{and, } \mathbf{b}^\perp(\mathbf{x}) = \begin{cases} \dfrac{[b_2(\mathbf{x}), -b_1(\mathbf{x})]}{|\mathbf{b}(\mathbf{x})|} & \text{if } |\mathbf{b}(\mathbf{x})| \neq 0 \\ 0 & \text{else } |\mathbf{b}(\mathbf{x})| = 0. \end{cases}$$

Here, $u_h$ represents the numerical solution obtained using the predicted $\tau_K$. The function $q(s)$ corresponds to the cross-wind derivative term, which plays a crucial role in limiting the smearing effect in the numerical solution.

The ConvStabNet architecture consists of three one-dimensional convolutional layers with sizes [64, 32, 32] respectively. Each layer employs a convolutional filter with a stride of one. The network takes as input the diffusion coefficient ($\epsilon$), convective velocity ($b_1(x,y), b_2(x,y)$), mesh size ($h$), and the norm of the gradient of the solution. The network is implemented from scratch using the PyTorch Paszke et al. (2019) and FEniCS A. Logg (2012); Logg & Wells (2010) libraries.

In Algorithm 1, we augment the cell-wise input by including the gradient of the norm of the numerical solution and predict the cell-wise $\tau$ values for the entire domain. During each epoch, we check if the strong form residual of the equation is below the threshold $\beta_{thres}$. If the residual exceeds the threshold, the training process is halted.

### 3.1 IMPACT OF CROSSWIND DERIVATIVE TERM IN THE LOSS FUNCTION

One significant addition in the cost function is the utilization of an a posteriori error indicator term, which includes the crosswind derivative term, in the loss function. This term plays a crucial role in controlling the smearing effect in the solution. Figure ?? illustrates the impact of including the crosswind derivative term on the $L^2$-error in the numerical solution obtained by minimizing the loss function. It is evident that incorporating the crosswind derivative term leads to a noticeable reduction in the $L^2$-error for all the considered examples.

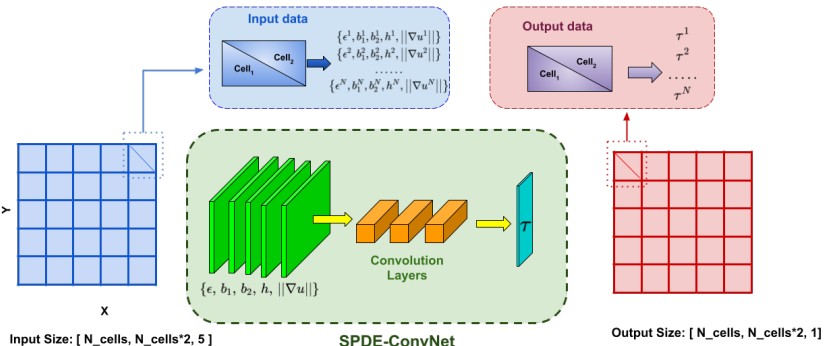

Figure 1: Network Architecture of ConvStabNet.

---

**Algorithm 1** ConvStabNet algorithm

---

1: Initialize learning rate $\eta_0$, $n_{epochs}$, $I^K = \{(\epsilon^K, b_1^K, b_2^K, h^K, ||\nabla u_h^K||_{0,K})\}$
2: Initialize the weights, $\theta_0$, of ConvStabNet with random values
3: Initialize the optimizer (Adam in this case) and stepLR scheduler
4: Solve equation equation 6 to get $u_h$ and $||\nabla u_h^K||_{0,K}$. Add $||\nabla u_h^K||_{0,K}$ to $I^K$.
5: **for** $t = 0$ to $n_{epochs}$ **do**
6:     $\tau_K(\theta_t) = \text{ConvStabNet}_{\theta_t}(I^K)$
7:     $\eta_t = stepLR(t)$
8:     Solve equation equation 8 with $\tau_K$ to get $u_h$
9:     $\text{Loss}(u_h(\theta)) = \sum_{K\in\Omega}(|| - \epsilon\Delta u_h + \mathbf{b}\cdot\nabla u_h - f||_{0,K}^2 + ||q(|\mathbf{b}^\perp\cdot\nabla u_h|)||_{0,1,K})$
10:     **if** $\left(\beta_h = \sum_{K\in\Omega}|| - \epsilon\Delta u_h + \mathbf{b}\cdot\nabla u_h - f||_{0,K}^2\right) < \beta_{thres}$ **then**
11:         break
12:     **else**
13:         $\text{loss}(\theta_t) = \beta_h + \sum_{K\in\Omega}||q(|\mathbf{b}^\perp\cdot\nabla u_h|)||_{0,1,K}$
        Backpropogate: $\theta_{t+1} = \theta_t - \eta_t\nabla_{\theta_t}\text{loss}(\theta_t)$
14:     **end if**
15: **end for**

---

## 4 NUMERICAL EXPERIMENTS

To assess the robustness of ConvStabNet, we conduct experiments using various examples. In this section, we provide the details of these examples, which are outlined below:

**Example 1: Constant source term**
We consider the equation equation 4 with following data:

$$\epsilon = 10^{-8}, \quad \mathbf{b} = (1,0), \quad f = 1, \quad \Omega = (0,1)^2, \ g = 0.$$

This example is taken from John & Knobloch (2007). The solution has an exponential layer at $x = 1$ and two parabolic layers at $y = 0$ and $y = 1$ respectively. The source function is constant and the analytical solution for this problem to test the performance of the proposed technique on this problem.

**Example 2: Variable source term**
We consider the convection-diffusion equation equation 4 with following equation coefficients and boundary conditions as given in Knobloch (2009):

$$\epsilon = 10^{-8}, \quad \mathbf{b} = (2,3), \quad g = 0.$$

The source term $f$ depends on x, y and is calculated by substituting the following analytical solution

$$u(x,y) = xy^2 - x\exp\left(\frac{3(y-1)}{\epsilon}\right) - y^2\exp\left(\frac{2(x-1)}{\epsilon}\right) + \exp\left(\frac{2(x-1) + 3(y-1)}{\epsilon}\right).$$

This solution contains two outflow boundary layers. one at $x = 1.0$ and another at $y = 1.0$.

**Example 3: No source term**

Next, we consider the convection-diffusion equation equation 4 with the following coefficients and boundary conditions:

$$\epsilon = 10^{-8},\ \theta = -\pi/3,\ \mathbf{b} = (\cos(\theta), \sin(\theta)),\ f = 0,$$

$$g = \begin{cases} 0, & \text{for } x = 1 \text{ or } y \leq 0.7 \\ 1, & \text{otherwise.} \end{cases}$$

This solution contains both exponential and boundary layers and is taken from John et al. (2011).

**Example 4: Discontinuous source term**

We consider the convection-diffusion equation equation 4 with the following coefficients and boundary conditions:

$$\epsilon = 10^{-8},\ \mathbf{b} = (1, 0),\ g = 0,$$

$$f = \begin{cases} 0, & \text{if } |x - 0.5| \geq 0.25 \cup |y - 0.5| \geq 0.25 \\ -32(x - 0.5), & \text{otherwise,} \end{cases}$$

$$u = \begin{cases} 0, & \text{if } |x - 0.5| \geq 0.25 \cup |y - 0.5| \geq 0.25, \\ -16(x - 0.25)(y - 0.75), & \text{otherwise.} \end{cases}$$

This example differs from Example 1 in the source function $f$. It has previously been employed in Knobloch (2008) as a benchmark case. In this scenario, the solution exhibits two interior characteristic layers in the convection direction, specifically between the spatial points $(0.25,\ 0.25)$ and $(0.25,\ 0.75)$. The Peclet number associated with this example is $1.77 \times 10^6$.

**Example 5: Variable convective velocity**

The proposed ConvStabNet is designed to handle SPPDEs with variable coefficients by incorporating the local convective velocity as an input. In order to validate its effectiveness, we consider the convection term as a spatially varying function:

$$\epsilon = 10^{-8}, \quad \mathbf{b} = (-y, x)^T, f = 0, \quad \Omega = (0, 1)^2,$$

$$g = \begin{cases} 1 & \text{if } \frac{1}{3} \leq x \leq \frac{2}{3} \text{ and } y = 0, \\ 0 & \text{else.} \end{cases}$$

### 4.1  MESH CONVERGENCE ANALYSIS

In FEM, the accurate simulation of complex physical phenomena and the attainment of reliable numerical solutions heavily rely on mesh resolution. Mesh refinement involves dividing the computational domain into smaller cells to capture finer details of the problem being studied. Increasing the number of cells in the areas of interest, such as regions with steep gradients or high-stress concentrations, enhances the accuracy of the solution. A refined mesh allows for a more precise representation of geometric features, boundary conditions, and material behavior, resulting in more accurate predictions of the system's response. However, mesh refinement comes at the expense of increased computational requirements since a larger number of cells need to be handled. Therefore, it is crucial to strike a balance between mesh size and computational efficiency to ensure accurate results without excessive computational overhead. In the case of *ConvStabNet*, we conducted a mesh refinement analysis, and the results are summarized in Table 1. It shows we have obtained optimal order of convergence.

Table 1: Errors in the numerical solution of Example 2

| $N_{cells}$ | h | Residual | $L^2$-error | Relative $l^2$-error | $H^1$-Seminorm | $L^\infty$-error | Order |
|---|---|---|---|---|---|---|---|
| 10 | 1.41e-1 | 8.71e-1 | 4.75e-4 | 3.83e-1 | 1.06e-2 | 1.04e-3 | |
| 20 | 7.07e-2 | 5.12e-1 | 2.93e-5 | 9.28e-2 | 1.34e-3 | 5.92e-5 | 4.02 |
| 40 | 3.54e-2 | 2.88e-1 | 3.63e-6 | 4.60e-2 | 3.29e-4 | 7.35e-6 | 3.01 |
| 80 | 1.77e-2 | 1.76e-1 | 4.51e-7 | 2.29e-2 | 8.14e-5 | 9.20e-7 | 3.01 |

## 4.2 THE CHOICE OF FINITE ELEMENTS

In addition to mesh refinement, the accuracy of the solution can also be improved by increasing the order of the finite elements. Opting for elements with higher polynomial order allows for a more precise representation of complex geometries and smooth solutions. However, this improvement often comes at the cost of increased computational requirements and a larger number of degrees of freedom to solve for. Additionally, higher order finite elements may introduce oscillations near interior and boundary layers.Therefore, it is crucial to carefully consider the behavior of the selected finite elements under different conditions and ensure they are suitable for capturing the relevant physics of the problem at hand. This involves assessing their ability to accurately represent the anticipated solution characteristics and evaluating their performance in capturing important features such as discontinuities, gradients, and boundary conditions. By selecting appropriate finite elements, one can strike a balance between accuracy and computational efficiency while ensuring the faithful representation of the underlying physics. In table 2, we show the performance of ConvStabNet for different pairs of finite elements, in terms of all error metrics. We found that the best pair is $(P_2, DG_0)$ for the considered example as all the error metrics are least for $(P_2, DG_0)$ pair of choice.

Table 2: Choice of finite elements for $u_h$ and $\tau_K$

| Pair of elements $(\tau_K, u_h)$ | Residual | $L^2$-error | Relative $l^2$-error | $H^1$ error | $L_\infty Error$ |
|---|---|---|---|---|---|
| $DG_0, P_1$ | 3.76e+0 | 9.35e-6 | 9.56e-2 | 8.72e-4 | 7.32e-5 |
| $DG_1, P_1$ | 2.54e+0 | 7.43e-6 | 8.53e-2 | 7.54e-4 | 5.45e-5 |
| $DG_1, P_2$ | 1.78e+0 | 5.34e-6 | 4.88e-2 | 5.32e-4 | 3.21e-5 |
| $DG_0, P_2$ | 2.62e-1 | 1.51e-6 | 1.27e-2 | 1.23e-4 | 1.23e-5 |

## 4.3 PERFORMANCE ANALYSIS

In order to assess and compare the effectiveness of ConvStabNet with VarNet, standard $\tau_{std}$, and AIStab-FEM, a comprehensive analysis is conducted, and the results are presented in Table 3. The error metrics associated with *ConvStabNet* were examined, revealing consistent outperformance compared to VarNet and standard $\tau_{std}$. This significant improvement in performance demonstrates the superiority of *ConvStabNet* over the other two techniques.

Table 3: Comparison of ConvStabNet with other techniques for the Example 2.

| | $L^2$-error | Relative $l^2$-error | $H^1$-seminorm | $L^\infty$ error |
|---|---|---|---|---|
| Standard $\tau$ | 6.77e-6 | 1.36e-1 | 6.74e-4 | 7.29e-5 |
| VarNet | 2.37e-4 | 1.62e+0 | 1.87e-3 | 3.55e-4 |
| *AI-stab FEM* | 5.04e-6 | 9.73e-2 | 4.80e-4 | 4.05e-5 |
| *ConvStabNet* | 3.04e-6 | 8.36e-2 | 3.20e-4 | 4.03e-5 |

## 4.4 QUALITATIVE COMPARISON

To comprehensively evaluate the performance of ConvStabNet and AIStab-FEM, the first four examples are considered. Figures 3 and 4 visually represent the predicted values of $\tau_K$ obtained from ConvStabNet and AIStab-FEM, respectively, for each of the examples. Notably, the predictions derived from ConvStabNet exhibit intricate local details, enhancing the fidelity of the results.

To exemplify the efficacy of ConvStabNet in handling equations with variable coefficients, we specifically present the solutions obtained for Example 5, in Figure 2. This figure demonstrates ConvStabNet's ability to accurately capture the varying coefficients and provide precise predictions. By observing the figure, it becomes evident that ConvStabNet effectively handles the complexities introduced by variable coefficients, showcasing its versatility and robustness across different scenarios. The difference between the $\tau_K$ predicted from AIStab-FEM and ConvStabNet is shown in Fig. 5, and we can see that ConvStabNet improves the predictions by a significant margin.

## 5 SUMMARY

In this study, we introduce ConvStabNet, a convolutional neural network designed to predict stabilization parameters for solving two-dimensional SPPDEs using the SUPG technique. The network

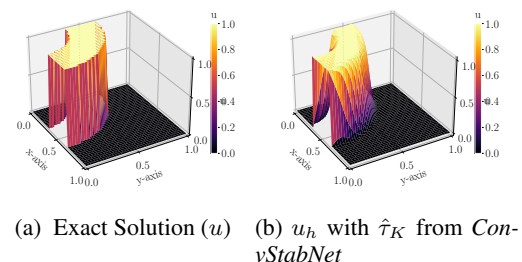

(a) Exact Solution ($u$)  (b) $u_h$ with $\hat{\tau}_K$ from *Con-vStabNet*

Figure 2: ConvStabNet for equation with variable coefficients (convection velocity).

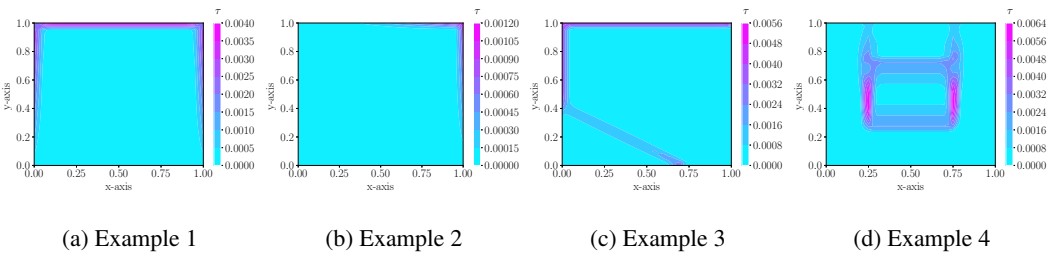

(a) Example 1  (b) Example 2  (c) Example 3  (d) Example 4

Figure 3: $\hat{\tau}_K$ predicted from *ConvStabNet*

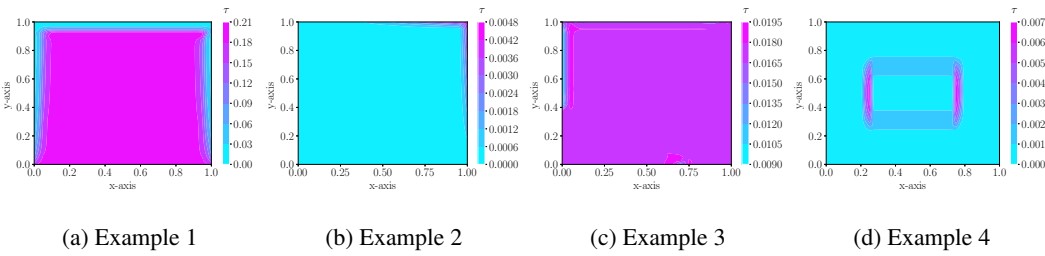

(a) Example 1  (b) Example 2  (c) Example 3  (d) Example 4

Figure 4: $\hat{\tau}$ predicted from *AIStab-FEM*

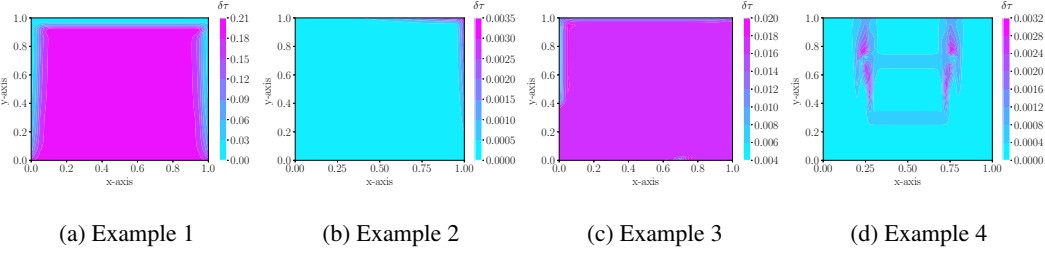

(a) Example 1  (b) Example 2  (c) Example 3  (d) Example 4

Figure 5: Difference of $\hat{\tau}_K$ predicted from *AIStab-FEM* and *ConvStabNet*

utilizes localized cell features to predict cell-wise $\tau_K$. To assess the performance of ConvStabNet, we conducted several experiments, including mesh refinement analysis and generalizability testing. The proposed technique was compared against other approaches, such as Std. $\tau_{std}$, *VarNet*, and AIStab-FEM in terms of various evaluation metrics including Residual, $L^2$ Error, Relative $l^2$-error, $H^1$ Error, and $L_\infty$ Error. The results demonstrate that *ConvStabNet* outperforms all of these methods. Furthermore, *ConvStabNet* addresses the limitations of *AIStab-FEM* Yadav & Ganesan (2022) by showcasing good performance even in scenarios where equation coefficients vary. This highlights the robustness and versatility of the proposed approach in handling variable equation coefficients.

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
