# OpenReview forum: "OPTIMIZING STABILIZATION IN SINGULARLY PER- TURBED PROBLEMS WITH SUPG SCHEME"
_ICLR.cc/2024/Conference — ICLR 2024 Conference Withdrawn Submission_

### Official Review · Reviewer_WvoG · 2023-10-24

**Soundness:** 1 poor
**Presentation:** 3 good
**Contribution:** 1 poor
**Rating:** 3
**Confidence:** 5

**Summary:**

The submission studies a CNN-based strategy to "learn" the stablization paramater for the Streamline Upwind Petrov-Galerkin method for convection-dominated problems. Due to flawed philosophy to design the method in the first place, I cannot recommend acception.

**Strengths:**

- Convection-dominated convection-diffusion equation is important in many problems (MHD, EM, fluid, etc), and its accurate numerical approximation still poses many open problems in the scientific computing community. This paper studies one of them, the optimal stabilization parameter $\tau_K$ in 2D for the SUPG discretization of the convection-dominated problem.
- The use of $\mathbf{b}\_K$ and $\\|\nabla u_h\\|_{0,K}$ as different input channels of CNN is a good practice (though the references are missing).

**Weaknesses:**

There are several major weaknesses, and all of them, most likely, attribute to the authors' misunderstanding of the cause of the problem of "spurious oscillations" for numerical methods for convection-dominated problems, as well as what SUPG is trying to do (or rather to say, what compromises, in terms of consistency vs stability, does SUPG make to get the ball rolling).

- For example, on page 3, it says "A small value of $\epsilon$ often induces spurious oscillations in the standard Galerkin solution.", this is wrong. If the mesh is sufficiently fine, or the mesh can resolve the crosswind in a certain way, there will be no spurious oscillations for even standard Galerkin approximations. Please see all those studies in the anisotropic meshing for convection-dominated problems.
- Continuing the point above, why meshing is more important than $\tau_K$ is simply because on a given mesh (that is not fine enough to resolve the layers), $\tau_K$ cannot be chosen to resolve the interior layer perfectly with no oscillation, and at the same time gives a good non-over-smoothened solution. For example, the plot of $u_h$ in Figure 2 looks like the typical over-smoothened ones.
- One reason that the oscillation is present is because the numerical solution does not satisfy the (discrete) maximum principle (or not monotone per se).  The maximum principle guarantees no interior maximum (thus no oscillation) is achieved on either side of the characteristics (where the sharp layers are located). The authors did not acknowledge all the developments of monotone methods for convection-dominated problems. For example, a Scharfetter–Gummel type exponential fitting scheme for the linear finite element can eliminate the oscillations completely (on a mesh SUPG either over-stablizes and cannot resolve the interior layer, or under-stablizes thus oscillatory). Please check the figures in Xu-Zikatanov 1999 Math. Comp. paper to see what a non-oscillatory non-over-smoothened numerical solution is like.
- The authors claimed the standard SUPG stabilization parameter "lacks extendability to high-dimensional problems", in fact, people in this field have known the criterion in higher dimensions for a long time.  You want a decent amount of diffusion in the direction of streamlines/characteristics to make the coercivity robust in the $\epsilon$-weighted SUPG norm (so you want it sufficiently big), but without making too much compromise in the (local) consistency (so you want it as small as possible). The optimal choice should in fact the solution to a saddle-point problem (constraint minimization) locally, ***NOT globally*** like the ones obtained using Alg 1 line 9 in this submission. For example, see A. Russo 1996 CMAME paper which analyzes what condition the stabilizing bubble should satisfy. This point brings the next point.
- The loss function is said to be "inspired" by the error indicators in John-Knobloch-Savescu 2011 CMAME paper. However, in that paper, $\tau_K$ is solved through the critical point argument of the error functional (solution to an adjoint variational problem in $L^{\infty}$, an analogy in optimization would be the convex conjugate to a saddle point problem). Not a direct minimization.

### Minor ones

Here are the list of minor weaknesses such as typos, unidiomatic notations, wrong claims.

- All the equation references seems to have an extra "equation" due to some macro setup, for example, on page 2: "In Equation equation 1, ...", "Lossequation 3", and "In Equation equation 3, ...", and this error carries on to later pages as well.
- The authors give the problem of interest, the Singularly Perturbed Partial Differential Equations, the acronym the SPPDEs. In my opinion, this is misleading, because there is singularly perturbed reaction-diffusion equation (which does not need SUPG). The keyword here should be "convection-dominated", i.e., convection-dominated convection–diffusion equations.
- On Page 4, if $\Omega_h$ is a partition of $\Omega$ (triangulation, Cartesian grid, polytopal, etc), then $H^1(\Omega_h)$ in many occasions represents the *piecewise* $H^1$ space with respect to this partition, i.e., $H^1(\Omega_h):= \Pi_{K\in \Omega_h} H^1(K)$, not a finite dimensional subspace. For example, please see the Arnold-Brezzi-Cockburn-Marini 2002 SINUM paper (about discontinuous Galerkin methods).
- Equation (8) on page 4, if the authors ought to write the weak formulation in the Galerkin form, then the part involving $\tau_K(f_h, \mathbf{b}\cdot\nabla v_h)_K$ conventionally appears on the right hand side, because $a^{\text{SUPG}}(\cdot,\cdot)$ is a bilinear form for $u_h$ and $v_h$ and this term is not a part of any bilinear form (see e.g., John-Knobloch-Savescu 2011 CMAME paper in the reference). Alternatively, the left hand side of the equation is sometimes written as $a(u_h, v_h) + \sum_K (R_K(u_h), \tau_K \mathbf{b}\cdot\nabla v_h)_K$ where $a(u_h, v_h)$ is the usual Galerkin bilinear form (see e.g., John-Knobloch 2007 CMAME paper in the reference).
- Page 4, if the authors meant to represent $f$ as discretized in the context of computing the integral quadrature, then both $f$'s in (6), (8) should be $f_h$. Right now they are mixed together.
- Page 5, the author claims "the function $q(s)$ corresponds to the cross-wind derivative term", this is imply wrong. The cross-wind directional derivative is $\mathbf{b}^{\perp}\cdot\nabla u_h$. Judging by the form of $q(\cdot)$, it is a common tool that guarantees that the cross-wind derivative is still differentiable at 0 (due to the presence of the absolute value) in the context of John-Knobloch-Savescu 2011 CMAME paper.
- Page 4, the local Péclet number should have $h_K$ not $h$.
- Sometimes, the flow field is denoted using boldface $\mathbf{b}$, while on some other occasions it is just $b$ (equation (7)).
- Sometimes, the mesh size is denoted by $h_K$ (section 2.3), while on some other occasions it is $h^K$.
- Personally, I would not use "advection" and "convection" to describe the same term in the same paper.

**Questions:**

The questions following are more like weaknesses, however, when re-submitting to a conventional computational physics/math venue, it is suggested that the authors adding these in the reports, as well as fixing the erroneous claims mentioned above (mostly about the finite element methods for convection-dominated problems). A good proof-read from a person familiar with finite element methods for convection-dominated problem is helpful as well.

- The key quantity in the loss function, $\\|\cdot\\|_{0,1,K}$ norm goes undefined.
- While it is okay to report $L^2$-error, $L^{\infty}$-error, and $H^1$-error, it is the tradition to report the error associated with the bilinear norm (aka SUPG norm).
- Page 7 Table 1, if the mesh size halves, the number of cells should quadrupled in 2D. The table is wrong. The order of convergence does not match any of the numbers in the table either.

---

### Official Review · Reviewer_pFUX · 2023-10-29

**Soundness:** 3 good
**Presentation:** 3 good
**Contribution:** 2 fair
**Rating:** 5
**Confidence:** 3

**Summary:**

The paper introduces a novel approach to address the stabilization issue that is raised when solving perturbed partial derivative equations (PPDEs) using some physic informed neural networks (PINNs). While the current approaches predict the stabilization parameter globally, the authors suggest to learn a parameter that take different values for each cell of the 2D grid. To do so, the authors use a convolutional network that is learned using an iterative procedure. The paper presents some numerical experiments showing the efficiency of their approach against some approaches from the literature.

**Strengths:**

The paper introduces a method that addresses a specific issue raised in the resolutions of  PPDEs using ML methods. The learning methodology is well explained and the numerical experiments show that the resulting error obtained is lower than the other PINNs-based approaches.

**Weaknesses:**

In my opinion, the paper has several major weaknesses.

Major comments:

   - Although the efficiency of the overall approach is proved on several numerical experiments, the paper attempts to address an issue that seems to be very specific, which makes the contribution of the paper relatively low compared to ICLR standards. In my opinion, the positioning of the paper would make it more suitable for submission to physics journal. To be clear, the contribution is relevant for the domain, but the contribution to the ML aspect is not significant.

   - The cross-wind derivative term seems to play an important role in the efficiency of the approach. However, there is a lack of clarity on its meaning for a non-expert reader. The authors claim that this is a crucial point without providing the intuition behind its use.

   - One of the main weakness of the paper is the numerical experiment section. The authors introduce several examples from the literature, but they report the error for only one of them (Example 2). The qualitative comparison is also difficult to understand. The authors claim that the approach captures varying coefficient and local details, "enhancing the fidelity of the results". What does it mean exactly? Is this learned stabilization parameter theoretically approved? In my opinion, capturing local details does not necessarily mean that the reconstructed solution has a better accuracy.

Minor comments:

   - In the introduction, the authors explain that the value of the stabilization parameter has to be carefully chosen. Could the authors provide more details about the trade-off? Why do we want to keep a reasonable amplitude of oscillations?

  - There is a compiling problem at the end of the numerical experiments section.

**Questions:**

See comments above.

 - Is the code publicly available?

Depending on authors responses, I would change my score.

---

### Official Review · Reviewer_ptK7 · 2023-11-01

**Soundness:** 3 good
**Presentation:** 3 good
**Contribution:** 2 fair
**Rating:** 3
**Confidence:** 5

**Summary:**

Numerical solution of advection-dominated PDEs, such as Singularly Perturbed Partial Differential Equations (SPPDEs) is a nontrivial task. Galerkin methods alone may produce oscillations or spurious solutions. The SUPG method introduces a stabilizing term that takes into account the flow direction, effectively handling the convection effects and improving the accuracy of the solution. Comparison of equations 6 and 8 clarifies the impact of added term, which is proportional to cell-wise \tau_K. The solution is sensitive to such stabilization parameter to avoid over-damping (large values) or numerical stability (small values). Such stabilization parameter field is learned via convolutional network, to account for 2D fields as input and retain the connectivity/neighborhood information. Algorithm 1, along with figure 1 clarifies the method, along with the choice of parameters, input, output and more importantly the choice of loss function. Various examples are solved and compared with SOTA (based on equation 9) in section 4.

**Strengths:**

The paper is well written. The numerical method is valid and the paper can sit well in the computational PDE community. Comparison with one-dimensional approximation is an appropriate baseline. Examples are in the context of advection dominated PDE class and relevant. Weak form formulation is introduced properly, given the space of the paper. The choice of loss function is meaningful and nontrivial.

**Weaknesses:**

My major concern of this paper is the scope. Even in the context of stabilizing the weak formulation for CFD applications one may think of alternative methods such as Petrov-Galerkin (PG) or its variants like Discontinuous Galerkin or cross-wind  stabilization methods. Authors can argue that for the specific advection dominated PDEs, SUPG may be superior but this not a very major contribution. This paper can be good candidate for numerical journals (and even then authors may want to solve more complicated examples or geometries; the current examples are really not very challenging) but due to its limited scope and not involving examples have a limited scope of applications.
I also have some questions which I'll refer to below section.

**Questions:**

Inspired by variational PINNs and the use of SUPG is an interesting problem formulation. But could authors formulate the problem as parameter-dependent PINNs, such as the ones arise in inverse problems: the loss function includes the governing equation plus a regularization term (which could be of the form of residual introduced in equation 7) and the input is collocation points (spatial coordinates) plus \tau_K? Authors don't need to actually solve such alternative formulation but it would be nice to have a conversation about this.

Alternatively, can one alter the architecture of the paper and use the trained network as the solution of PDE itself? This may be done by having u in addition to \tau_K as output (loss function may need modification too).

There are various typos in the paper; for instance at page 5 Figure ?? is shown which should be modified in the original latex. Also, at the end pf page 8, the text is not legible. Please improve the quality of the paper in the revised version.